

# Hemispheric asymmetry in stratospheric NO₂ trends

Margarita Yela[1], Manuel Gil-Ojeda[1], Mónica Navarro-Comas[1], David Gonzalez-Bartolomé[1], Olga Puentedura[1], Bernd Funke[2], Javier Iglesias[1], Santiago Rodríguez[1], Omaira García[3], Héctor Ochoa[4] and Guillermo Deferrari[5, 6]

[1]Atmospheric Research and Instrumentation Branch. Instituto Nacional de Técnica Aeroespacial (INTA), Ctra. Ajalvir s/n, Torrejón de Ardoz, 28850 Madrid, Spain.
[2]Instituto de Astrofísica de Andalucía (CSIC), Granada, Spain.
[3]Centro de Investigación Atmosférica de Izaña (CIAI), Agencia Estatal de Meteorología (AEMET), Spain.
[4]Dirección Nacional del Antártico/Instituto Antártico Argentino, 25 de Mayo 1143, San Martín Provincia de Buenos Aires, Argentina.
[5] Centro Austral de Investigaciones Científicas (CADIC), Ushuaia, Argentina.
[6] Universidad Nacional de Tierra del Fuego (UNTDF), Ushuaia, Argentina.

*Correspondence to*: Margarita Yela (yelam@inta.es)

**Abstract.** Over twenty years of stratospheric NO₂ vertical column density (VCD) data from ground-based zenith DOAS spectrometers was used for trend analysis, specifically, via multiple linear regression. Spectrometers from the Network for the Detection of Atmospheric Composition Change (NDACC) cover the subtropical latitudes in the Northern Hemisphere (Izaña, 28°N), southern Subantarctica (Ushuaia, 55°S) and Antarctica (Marambio, 64°S and Belgrano, 78°S). The results show that for the period 1993-2014, a mean positive decadal trend of +8.7% was found in the subtropical Northern Hemisphere stations, and negative decadal trends of -9.1% and -13.8% were found in the Southern Hemisphere at Ushuaia and Marambio, respectively; all trends are statistically significant at 95%. Belgrano only shows a significant decadal trend of -11.3% in the summer/autumn period. Most of the trends result from variations after 2005. The trend in the diurnal build up per hour (DBU) was used to estimate the change in the rate of N₂O₅ conversion to NO₂ during the day. With minor differences, the results reproduce those obtained for NO₂. The trends computed for individual months show large month-to-month variability. At Izaña, the maximum occurs in December (+13.1%), dropping abruptly to lower values in the first part of the year. In the Southern Hemisphere, the polar vortex dominates the monthly distributions of the trends. At Marambio, the maximum occurs in the mid-winter (-21%), whereas at the same time, the Ushuaia trend is close to its annual minimum (-7%). The large difference in the trends at these two relatively close stations suggests a vortex shift towards the Atlantic/South American area over the past few years. Finally, the hemispheric asymmetry obtained in this work is discussed in the frame of the results obtained by previous works that considered tracer analysis and Brewer-Dobson circulation. The results obtained here provide evidence that the NO₂ produced by N₂O decomposition is not the only cause of the observed trend in the stratosphere and support recent publications pointing to a dynamical redistribution starting in the past decade.





## 1 Introduction

Stratospheric nitrogen dioxide ($NO_2$) has been a subject of interest since the late sixties, when the use of supersonic aircraft flying at stratospheric levels was being considered. At that time, studies on the impact of nitrogen oxides on the ozone in the
stratosphere were initiated (Johnston, 1971). The interest in this atmospheric layer was enhanced in the mid-seventies, powered by the discovery of the ozone depletion potential of halogenated compounds (Stolarski and Cicerone, 1974).

Nitrogen oxides interact with ozone both directly and indirectly. Nitric oxide (NO) reacts with ozone, forming $NO_2$ and $O_2$. In the daytime, $NO_2$ is rapidly photolyzed, and NO is recovered. This catalytic reaction results in ozone reduction. On the other hand, $NO_2$ reacts with BrO and ClO, mitigating the ozone depletion potential of halogenated compounds. $NO_x$
oxidation products, such as $HNO_3$ and $N_2O_5$, have relatively long lifetimes and therefore act as $NO_2$ reservoirs.

Regular monitoring of stratospheric $NO_2$ started in the early eighties with the deployment of Zenith DOAS (Differential Optical Absorption Spectroscopy) scanning spectrometers at Antarctic stations (Scott Base at 78° S by Mckenzie and Johnston, 1984 and Dumont D'Urville at 66° S by Pommereau and Goutail, 1988). In the following decades, instruments were installed at remote, unpolluted sites for monitoring $NO_2$, $O_3$ and other stratospheric gases suitable for ground-based
measurements.

Recently, the interest in trends of stratospheric trace gases has increased, since global circulations models predict an acceleration of the stratospheric residual circulation (or Brewer-Dobson circulation, BDC hereafter) of 2-3.2 %/decade as a consequence of the temperature changes in the atmosphere due to human activities (Butchart, 2014). The speed of the BDC circulation influences the concentrations of long-lived trace gases with sources in the troposphere (e.g., CFCs, $CH_4$, $N_2O$), as
well as their eventual stratospheric products (Cook and Roscoe, 2009). In particular, $N_2O$, which is the primary source of $NO_x$, dissociates in the stratosphere via a reaction with excited atomic oxygen ($^1D$) and produces NO. $NO_2$ is mainly formed by the $NO+O_3$ reaction. An accelerated BDC implies a lower mean residence time of $N_2O$ in the stratosphere and, consequently, a lower $N_2O$ to $NO_2$ conversion.

On the other hand, the emissions of $N_2O$ to the atmosphere are steadily growing, with the concentration increasing
at a rate of 2.5 % / decade and more in recent years (IPCC, 2013). Since the $N_2O$ lifetime in the troposphere is approximately 120 years and $N_2O$ is the major source of $NO_2$, a tropospheric increase will result in an $NO_2$ increase in the stratosphere. These two counteracting processes could minimize or even cancel the potential changes in the stratospheric concentration of $NO_2$. However, the observational evidence of a trend in the BDC speed remains inconclusive (Butchart, 2014, Fu et al., 2015). Oberländer-Hayn et al., (2016) suggest that, even though the stratosphere is changing substantially in response to Greenhouse
Gases (GHG) increases, most of the BDC trend is associated with a lifting of the entire atmospheric circulation instead of an accelerating flow. If this is the case, a $NO_2$ increase should be seen in the long term, given that all other factors remain the same.

Previous studies to establish a trend in stratospheric $NO_2$ based on the DOAS data from the stations with long data availability, however, did not yield conclusive results on the global scale. A first attempt was undertaken by Liley et al.,



(2000) using the longest data series available, who found an approximate 5 % increase at Lauder (45°S) for the period 1981-1999. Fish et al., (2000) found no explanation related to the changes in $O_3$, temperature or water vapour, concluding that only changes in the stratospheric aerosol load could explain the observations. Later, McLinden et al., (2001) reproduced the observations of Liley et al. (2000) using a 3D CTM model, including halogen chemistry and assuming a negative trend in temperature of 0.5°/decade. A few years later, Gruzdev (2009) analysed 23 stations from the NDACC database. Trends were

found to be mostly positive in the middle and low latitudes of the Southern Hemisphere (SH) and mostly negative in the European sector of the middle latitudes of the Northern Hemisphere (NH). In the high and polar latitudes of both hemispheres, the annual estimates of the trends were mostly statistically insignificant. In Antarctica, a positive $NO_2$ trend was observed at 78° S, while in the NH high latitudes, both positive and negative trends were observed. A detailed study of the Jungfraujoch station (55° N, 3550 masl), including DOAS plus FTIR data and satellite composites (Hendrick et al., 2012), found negative

trends ranging from -2.4 % to -4.3 %, depending on the instrument and selected period. In summary, until now, the global $NO_2$ trend over the past few decades has not been accurately established. One important difficulty in comparing the trends at different latitudes is that the stations do not cover the same periods. In addition, the trends can be the result of multiple factors that do not behave linearly.

The purpose of this work is to contribute to the knowledge of the long-term $NO_2$ evolution by searching for significant
trends in the stratospheric $NO_2$ using the INTA $NO_2$ DOAS records in combination with a multiple regression model. Such models have been extensively used in recent years to infer trends in long-term atmospheric time series of ground-based instruments; in particular, they have been used to study the evolution of stratospheric ozone (Bodeker et al., 1998; Wohltmann et al., 2007; Fioletov 2008; Mäder et al., 2010), $NO_2$ (Van der A et al., 2006 and 2008; Gruzdev, 2009, Hendrick et al., 2012) and other species (i.e., Remsberg, 2015).

The paper is organized as follows: In section 2, the instruments and data are described. In section 3, the regression model and explanatory variables are presented. Section 4 addresses the results and is composed of 3 subsections. In 4.1, the general results are shown. Section 4.2 addresses a more detailed analysis of the subtropical case and, in section 4.3, the interhemispheric asymmetry is discussed. Finally, the conclusions are shown in section 5.

## 2 Instrumentation, stations, techniques and database

In the year 1993, INTA installed a scanning spectrometer at the subtropical high mountain observatory of Izaña (28° N, 16° W, 2370 masl) for long term measurements of stratospheric $NO_2$. In 1994, two identical instruments were deployed in Subantarctica (Ushuaia station; 55° S, 68° W) and Antarctic Peninsula (Marambio station; 64° S, 56° W). Then, in 1995, a fourth instrument was installed Antarctica at a higher latitude (Belgrano station; 78° S, 35° W). The three instruments cover the latitudinal belt from the outside to the inside of the Antarctic vortex. Since installation, all instruments have been operated

without interruption.



Belgrano (in continental Antarctica) is usually closest to the core of the polar vortex and hence is representative of the in-vortex air. Marambio (Antarctic Peninsula) is frequently located at the edge of the vortex region and alternates between measuring the vortex and the mid-latitude air masses. Ushuaia is, essentially, a mid-latitude station and only occasionally is reached by in-vortex air

The instrumentation consists of spectrometers covering the visible range, and the retrieval is based on the DOAS technique. At all four stations, identical scanning spectrometers (EVA) were initially installed. In Izaña, a second spectrometer (RASAS), based on a diode array detector, covering a wider range was added in 1999. Then, in 2010, the PDA was replaced by a CCD (RASAS-II), including MAXDOAS capability. All instruments were developed at the INTA laboratories. Data merging was carefully carried out after a period of overlap to assure the smooth transition between the instruments. The
transition between EVA and RASAS has been previously reported (Gil et al., 2008). The transition between RASAS and RASAS-II required a RASAS correction due to a degradation of the detector response since mid-2006, at a rate of 4.33 % / year. The details of the RASAS-II instrument have been previously published in Puentedura et al., (2012) and Robles-Gonzalez et at., (2016). The spectrometers were installed in the top terrace of the Izaña Atmospheric Observatoy, run by the CIAI (Centro de Investigación Atmosférica de Izaña), belonging to the Agencia Estatal de Meteorología (AEMET, Spain),
on the slopes of the Teide volcano, Tenerife, Canary Island. The Izaña Atmospheric Observatory is a high mountain station, part of the Global Atmospheric Watch (GAW) programme and managed by the CIAI.

The EVA instrument is a scanning spectrometer for twilight measurements between 88-92° solar zenith angles (SZA) in the 430-450 nm spectral range, with a spectral resolution of 1 nm. These instruments are located outdoors in thermostatic housings. The three Southern Hemisphere spectrometers were compared to each other after one year of measurements with
the help of a $NO_2$ cell containing a known amount of gas. The discrepancies among them were found to be below 4 % (Yela et al., 2005). Twilight (AM and PM) vertical column densities were derived from 6 typical individual measurements between an 88° and 91° SZA. New MAXDOAS instruments have been installed at the three stations, in 2011 at Belgrano, in 2015 at Marambio and in 2016 at Ushuaia. Both, DOAS and MAXDOAS instruments are simultaneously measuring for data serie homogenisation.

The $NO_2$ column retrieval is based on the standard DOAS spectral analysis (Platt and Stutz, 2008), performed using software developed at INTA. The DOAS settings for the $NO_2$ column retrieval follows the NDACC UV/Vis Working Group recommendations (Van Roozendael and Hendrick, 2012) whenever possible. Absorption cross sections $O_3$, $NO_2$, $H_2O$ and $O_4$ have been also included in the analysis. Raman scattering cross section was generated by the Win-DOAS package (Fayt and Van Roozendael, 2001), calculated from Raman theory. Finally, the inverse of the reference spectrum was included as a
pseudo cross section to account for stray light inside the spectrograph and the residual dark current of the detector. The air mass factor (AMF) used for the conversion of the $NO_2$ slant columns to vertical columns is the NDACC $NO_2$ standard AMF, available on the NDACC UV-Vis web page (http://ndacc-uvvis-wg.aeronomie.be/) and based on the Lambert et al., 1999 and 2000 climatology of the $NO_2$ profiles. This climatology consists of a Fourier harmonic decomposition of the UARS HALOE





v19 and SPOT-4 POAM-III v2 $NO_2$ profile data. The cross sections and other parameters used in the analysis are shown in

Table 1.

The estimated overall errors in the individual measurements are, on average, approximately 12 % (1-2 % fit analysis; < 2 % cross-sections; 2 % reference spectrum; 2-3 % AMF; 2 % stratospheric temperature). For details, see Gil et al., (2008). Fit analysis, cross-sections and reference spectra do not affect the trends. Potential long-term changes in the stratospheric temperature could have a minor effect on the cross-sections, but no statistically significant changes in the stratospheric

temperature have been observed during the data period. Only differences in the $NO_2$ profiles with respect to the AMF climatology used, could have an effect of a few tenths of a percent.

The spectrometers are NDACC-qualified instruments (more information is available at http://www.ndsc.ncep.noaa.gov/data/) and have been successfully intercompared for $NO_2$ in ad hoc international exercises (Roscoe et al., 1999 and 2010, Vandaele et al., 2005).

The DOAS technique at the zenith during twilight is slightly sensitive to clouds. Gaps in the data are mostly due to instrumental malfunctions. For monthly data, the rates of failures are 3.45 %, 0.40 % and 0.79 % for Izaña, Ushuaia and Marambio, respectively.

### 3 Multiple regression and proxies.

In the present work, a multiple linear regression model of the following form was used:


$$Y(t) = a + \sum_{k=1}^{3} \left[ \beta_{2k-1} \sin\left(2\pi t \frac{k}{12}\right) + \beta_{2k-1} \cos\left(2\pi t \frac{k}{12}\right) \right] + \sum_{j=7}^{m+4} \beta_j X_j(t) + \varepsilon(t) \qquad (1)$$

where,

$Y(t)$ = $NO_2$ column at time t

$a$ = constant

$X_j$ = each of the explanatory functions

$\beta_j$ = coefficient of the corresponding explanatory function $X_j$

$t$ = time since the start of the measurements

$\varepsilon$ = noise as a function of time $t$

The bracketed terms represent the harmonic functions accounting for the annual, semi-annual and quarterly waves (K=3), whereas the $\beta X$ terms include all explanatory variables. In the first run, 7 proxies that could potentially affect the $NO_2$ distribution are used (the trend, stratospheric aerosols, solar cycle, Quasi-Biennial Oscillation (QBO), stratospheric temperature, stratospheric circulation, and El Niño Southern Oscillation (ENSO)). Following Mäder et al., (2007), an iterative



process was used to exclude the proxies exceeding $\alpha > 0.1$, corresponding to significant values of less than 90%. For details, see Mäder et al., (2007) and Knibbe et al., (2014). In this way, the degrees of freedom of the regression increase.

Autocorrelation affects the linear trend calculation by increasing the uncertainty. If autocorrelation of the data noise is not included, the standard deviation of the trend estimate will substantially underestimate the actual uncertainty. According to Weatherhead et al., (1998), the standard deviation of the trend per year to be used to describe the precision of the trend

estimate can be approximated quite accurately by:

$$\sigma_\beta \approx \frac{\sigma_N}{n^{3/2}} \sqrt{\frac{1+\phi}{1-\phi}} \qquad\qquad (2)$$

where $\sigma_N$ is the standard deviation of the residuals (differences between the $NO_2$ data series and the modelled one), and n is the length of the data series in years and $\phi$ is the autocorrelation in the residual for time lag 1, defined as Corr ($N_t, N_{t-1}$). This formula has been previously used for trends in $NO_2$ (Van der A et al., 2006 and 2008, Hendrick et al., 2012).

**Stratospheric aerosols.** Stratospheric aerosols (SA) affect the amount of available $NO_x$. An increase in the aerosol loading due to volcanic eruptions reduces the $NO_x/NO_y$ partitioning through the heterogeneous hydrolysis of $N_2O_5$ on the aerosol surfaces (Fahey et al., 1993). The eruption of Mount Pinatubo in 1991 produced a reduction in the total $NO_2$ column of 19-34 %, depending on the station, in the following years (Gruzdev, 2014). Therefore, SA have been included as a proxy in the analysis. The dataset is the monthly aerosol optical depth (AOD) for the NH at 0.55 μm, compiled at NASA's Goddard

Institute for Space Studies (GISS). The dataset ending in September 2012 has been extended in time to April 2014 by the optical spectrograph and infrared imaging system (OSIRIS) data and to the end of 2014 by the white light optical particle counter (WOPC) data (Kremser et al., 2016). For the analysed period, the proxy removes the contribution of low $NO_2$ in the first few years of the time series, which would otherwise bias the trend.

**Solar cycle.** The solar radiation flux at 10.7 cm (2800 MHz) is an excellent indicator of solar activity. The data used

are from the Penticton Radio Observatory in British Columbia. Unlike many solar indices, it can be easily and reliably measured on a day-to-day basis from the Earth's surface in all types of weather. The 11-year solar cycle proxy is included to account for potential chemical changes due to variations in UV-radiation.

**QBO.** The Quasi Biennial Oscillation (QBO) affects the circulation in the lower stratosphere and hence the species distribution (Gray and Russell, 1999). Recently, QBO signatures in the variability of the middle to upper stratosphere $NO_2$

where detected (Liu et al., 2011). We employ a commonly used monthly index, built using the mean of the zonal winds averaged from 3 equatorial stations (Canton islands, Singapore and Gan/Maldives) compiled by the Berlin Free University for 50 and 10 hPa.

**ENSO**. The El Niño/Southern Oscillation (ENSO) has been found to influence the distribution of minor species in the tropical lower stratosphere up to 27 km (Randel et al., 2009). We have included as a proxy/predictor the multivariate

ENSO Index (MEI) based on the six main observed variables over the tropical Pacific: the sea-level pressure (P), zonal (U) and meridional (V) components of the surface wind, sea surface temperature (S), surface air temperature (A), and total




cloudiness fraction of the sky (C). Negative values of the MEI represent the cold ENSO phase (La Niña), while positive MEI values represent the warm ENSO phase (El Niño). For details, see Wolter and Timlim, (2011). A time lag of one to six months has been tested since the middle stratosphere may take time to respond to tropical sea surface temperature anomalies (Sioris
et al., 2014).

**Stratospheric temperature.** There is a growing consensus that the global temperatures in the stratosphere show a negative trend, i.e., Schwarzkopf and Ramaswamy, (2006); however, its magnitude is still under debate (Seidel et al., 2016). A recent study by Randel et al., (2016) based on satellite data shows a cooling trend of -0.1 to -0.2 K decade$^{-1}$ in the lower stratosphere and up to -0.5 to 0.6 K decade$^{-1}$ in the middle stratosphere for the period 1979-2015, but most of this trend is due
to the large decrease prior to 1995. After that date, there is no significant trend observed (Seidel et al., 2016).

Temperature variations in the stratosphere modify the $NO_2$ concentration by changing the reaction rates. To account for the potential trends related to this effect, the stratospheric temperatures over the stations extracted from the Interim European Centre for Medium-Range Weather Forecasts (ECMWF) Re-Analysis (ERA-Interim) (Dee et al., 2011) have been used as a proxy. Temperature data are extracted from the 0.25° x 0.25° grid at levels corresponding to the height of the $NO_2$
maximum at each station, as obtained by the $NO_2$ harmonic climatology based on HALOE v19 and POAM-II data (Lambert et al., 1999), that is, 10 hPa for Izaña, 20 hPa for Ushuaia and 30 hPa for Marambio and Belgrano. We used the monthly mean of the average of the 00:00 h and 12:00 h UT values as a proxy.

**Stratospheric circulation.** Changes in BDC induce changes in the concentration of stratospheric species with tropospheric origins. Recent climate studies have found that BDC will intensify with the increase in greenhouse gases
(Butchart, 2014). With a faster meridional circulation, $N_2O$ of tropospheric origin has a shorter lifetime in the stratosphere, and, as a consequence, less $N_2O$ would be oxidized to NO (Cook and Roscoe, 2009). To identify potential changes in the long-term $NO_2$ evolution associated with this effect, the Eddy heat flux (EHF) (v'T') at 100 hPa, averaged over 45°-75° S of the hemisphere, where the observational stations are located, has been included as a proxy in the multiple regression analysis. EHF is proportional to the vertical component of the Eliassen-Palm (EP) flux (Fosco and Salby, 1999, Salby and Callaghan,
2004), which has been found to have good correlation/anticorrelation with the extratropical/tropical $O_3$ (Randel et al., 2002, Weber et al., 2011). Data for the calculation of EHF were obtained from the ERA-Interim data, averaged for every month of the time series. The cumulative effects of the total $NO_x$/$NO_2$ concentration were approximated by considering a delayed response in the EHF time series as follows (Brunner et al., 2006):

$$EHF(t) = EHF(t-1)e^{\frac{\Delta t}{\tau}} + EHF(t) \qquad (3)$$

where EHF(t) is the data obtained from the ERA-Interim winds and temperatures and $\Delta t$ is the data time unit. For the Antarctic/Subantarctic stations $\tau$ was set to 12 for months from April to September and was set to 3 for the rest of the year. For Izaña subtropical station it is set to 3 for the whole year.

In Table 2, the source of each proxy is summarized





## 4 Results

### 4.1 General

The multiple regression fit is shown in Fig. 1 for Izaña, Ushuaia and Marambio. The model explains between 86 to 96 % of the observed variance in all cases. The dominant pattern is, as expected, the seasonal wave, explaining between 57 to 85 % of the variance, followed by the linear trend, explaining from 6 to 27 % of the variance. The rest of the explanatory variables have modest contributions. The residuals and the trend were plotted together to search for common structures that could represent anomalies not captured by the model (Fig. 2). The morning and evening residuals look very similar, showing the non-random character of the departures of the model. Cross-correlation between the stations shows independent residuals (correlation coefficients: IZO/USH = +0.01, IZO/MAR=-0.08, and USH/MAR=+0.13). The trend is significant at 99 %.

The results of the analysis show that the seasonal waves and trend are the major contributors to the variance, explaining 84.3/83.4 %, 91.0/92.2 % and 89.2/94.8 % of the variance for AM/PM at Izaña, Ushuaia and Marambio, respectively. As an example, Fig. 3 shows the individual contributions of the less relevant proxy terms (other than the seasonal and trend terms) for Izaña. They all maintain standard deviations below $5 \times 10^{13}$ molec.cm$^{-2}$, representing less than 2 % of the NO$_2$ data series mean value ($2.45 \times 10^{15}$ molec.cm$^{-2}$). The details of the trend analysis are summarized in Table 3.

To further explore the contributions of each proxy to the retrieved trend, a sensitivity test has been performed for data of the three stations. The results are shown in Fig. 4. Each data point represents the trend obtained when different proxies are excluded from the analysis. The first data point is the trend inferred considering all proxies from the monthly data series. The next six data points represent the trends when single proxies are excluded from the analysis, followed by the results obtained when excluding successively more proxies in order of their significance. Finally, the last data point represents the result of a simple linear regression. The results show that only stratospheric aerosols have a large impact at all three stations. Although the influence was restricted to the first years of the data (1993-1995), before the decay of the large nitrate aerosols load injected in the stratosphere following Pinatubo's eruption in 1991, this influence does affect the trends of the complete series by 14 to 22 %. Since the stratospheric aerosols reduce NO$_2$, when removing the aerosols proxy, the trend increases with a positive trend (NH) and decreases when the trend is negative (SH). The quarterly wave (not shown) and EHF have no impact or significance and were removed from the final analysis. The impact of the stratospheric temperature is also very small, probably because the impact is accounted for by the seasonal waves. The rest have minor impacts (<5%).

Next, we explored the trend sensitivity to the data series length by reducing the period both at the starting and ending months. For this exercise, we chose the Izaña PM series. The results (Fig. 5) show that the trend remains essentially unchanged if the data series is shortened by up to 5 years at the start and up to 4 at the end, providing confidence in the stability of the trend. It can also be seen that the trend is largest during the last decade (2003-2014).

The decadal trends obtained from the mean AM and PM values for Izaña, Ushuaia and Marambio are +8.7 ± 1.2 %, -9.1 ± 1.2 % and -13.7 ± 2.2 %, respectively. Most of the observed trends occurred in the last decade (after 2003-2006), as revealed by Fig. 2. At Izaña, the AM decadal trend is larger than the PM trend, +9.5 ± 1.2 % vs. +7.9 ± 1.1 %, respectively.





McLinden et al. (2001), using a chemical transport model (CTM) to explain the $NO_2$ trend observed at Lauder (Liley et al., 2000), found that the larger trends derived from the AM data are caused by decreasing $O_3$, which, in turn, reduces the rate of $NO_2 + O_3$, allowing for more $NO_2$ to be present the next morning. At Izaña, the different AM and PM trends cannot be explained by changes in $O_3$ since a non-significant decadal trend of $+0.5 \pm 1.2$ % is obtained for the period of 1999-2012 by the FTIR instrument (Vigouroux et al., 2015). However, McLinden et al., (2001) predicted a decrease in the $NO_2$ trend of approximately 1 %, when exclusively considering the increasing halogens in the stratosphere between 1980-2000. Data from the past few decades show the opposite trend. Equivalent effective stratospheric chlorine (EESC) gas concentrations in the mid-latitudes have declined by 15 % from their peak values in the year 2000. In particular, HCl, the main reservoir of inorganic chlorine, displays a negative decadal trend of $5.9 \pm 1.5$ % between 1997 and 2012. The same declining rate is observed for bromine between 2000 and 2012 (WMO, 2014). Both halogens should result in an increase in the $NO_2$ trend. Moreover, the halogen decline would modify the daytime $NO_x$ partition, since $NO_2$ would increase during the illuminated hours and decrease in darkness (McLinden et al., 2001), creating a larger trend at dawn than at dusk, as is observed. In the SH, the AM and PM decadal trends are closer, $-9.3 + 1.3$ % and $-8.9 \pm 1.3$ % for Ushuaia, $-13.8 \pm 2.1$% and $-13.6 \pm 1.9$% for Marambio, for AM and PM, respectively. At these stations, the decrease in halogens is probably compensated by an increase in $O_3$. Details of the trend analysis are summarized in Table 3.

At Belgrano, the trend has been obtained by averaging the decadal trends for the individual months, when data are available. Since DOAS requires SZAs close to 90°, only the Feb-April and Aug-Nov periods are suitable for measurements. For the sake of simplicity, we will refer to these periods as the summer and winter seasons, respectively. During the winter, the $NO_2$ column reaches values close to zero, and its trends are non-significant. The mean summer decadal trend is $-11.3 \pm 4.0$ % ($-10.9 \pm 4.1$ % and $-11.7 \pm 4.0$ % for AM and PM, respectively). In Fig. 6, these results are summarized.

An alternative way to analyse the trends of the twilight data is to consider the evolution of the $NO_2$ diurnal build-up per hour (DBU). The DBU is essentially due to the $N_2O_5$ photodissociation, which is dependent on temperature, $O_3$, and hours of the night and can be expressed as:

$$DBU = \frac{NO2_{PM} - NO2_{AM}}{24 - \Delta t} = \frac{NO2_{PM}}{24 - \Delta t}\left(1 - \frac{\sum_{base}^{toa}[W(z)R(z)]}{\sum_{base}^{toa}W(z)}\right) \qquad (4)$$

where $NO2_{AM}$ and $NO2_{PM}$ are the columns at sunrise and sunset, respectively, and $\Delta t$ is the monthly mean of the diurnal night hours. R is the AM-to-PM ratio at a given height, and $W(z)$ is a $NO_2$ climatological profile acting as weighting factor. R can be computed (Senne et al., 1996) from:

$$R(z) = \frac{NO2_{AM}(z)}{NO2_{PM}(z)} = e^{-2K(z)O3(z)\Delta t} \qquad (5)$$

where K is the $NO_2 + O_3$ reaction rate constant and $O_3(z)$ is the ozone concentration at height z.




At Izaña, the DBU obtained from Eq. (4) using the $O_3$ and temperature profiles from the local ozonesounding agree well with the observations (Fig. 7). The resulting decadal trend for the period is of +5.5 ± 1.6 %, which provides a hint of the trends in the reservoirs. In the Subantarctic/Antarctic stations, the DBU displays negative decadal trends of -8.4 ± 1.8 % and -13.8 ± 2.4 % for Ushuaia and Marambio, respectively. In Belgrano, the values are too low (winter) or the AM and PM data are too close (February) to obtain the DBU, except in March and April. The BDU decadal trends for these months are -13.0 ± 7.3 % and -10.6 ± 2.9 %, which is probably not representative of the mean annual trend. In summary, all SH stations exhibit a negative decadal trend in their DBUs, ranging from -8 to -14 %, and thus there was less $N_2O_5$ in the past few years at the middle and high latitudes of the Southern hemisphere.

To investigate the variability of the $NO_2$ trend with the season, the calculations were performed on a month-by-month basis. Even though the contributions of proxies other than seasonal waves are small (<1 % in most cases), the same proxies and lags were applied in the multiple regression as in the previous cases. The results are shown in Fig. 8. The AM and PM trends were plotted separately for each station, along with their mean value. At Izaña, during the winter and spring, both trends agree well. From July to the end of the year, as the trend increases, there is also a progressive increase in the AM-PM difference that clearly exceeds the error bars. $O_3$ trends in individual months (not shown) are too small and do not change much throughout the year. A seasonality in the decreasing halogen trend could be a possible explanation, but, for the moment, we have no satisfactory explanation for this behaviour. At Izaña, the monthly mean trends are largest in the autumn and beginning of the winter and are mainly forced by a faster increase in the AM trends. The annual excursion ranges from 4.4 % at its minimum in February to 13.1 % in December. The negative trend in the Southern Hemisphere increases towards the high latitudes from Ushuaia to Marambio; however, during the winter months, both stations exhibit opposite behaviours. At Ushuaia, the negative trend is reduced to values close to the annual minimum (-7 %), whereas in Marambio, the trend increases with respect to the previous months (-21 %). The time when the trends diverge coincides with the formation of the Antarctic Polar Vortex (APV), and this divergence extends for a period coincident with the coldest temperatures and polar stratospheric cloud (PSC) formation. In Fig. 8, the 2006 - 2016 mean area covered by temperatures low enough for PSC formation, as obtained by the Climate Prediction Center (CPC) (http://www.cpc.ncep.noaa.gov/products/stratosphere/polar/polar.shtml) is shown in grey. In September, the absolute maximum of the negative trends is reached at both stations as a result of an increase in the number of days that the stations were inside the APV. We will come back to this point later.

During the autumn and winter, there is meridional transport towards the Pole. Once the Antarctic vortex is established, $O_3$ and $NO_2$ transported from lower latitudes accumulate outside the vortex, since the vortex edge acts as a barrier for mass exchange. Marambio is generally at the edge of the vortex, whereas Ushuaia remains outside, with very few exceptions. The observed behaviour could be explained by a positive trend in the size of the vortex area, such that Marambio remains inside the vortex longer, thus reducing its $NO_2$ column year by year. Conversely, larger $NO_2$ masses transported from lower latitudes would accumulate above Ushuaia, somehow compensating for the negative trend. Alternatively, the same effect could be caused by a drift of the average vortex position towards the Atlantic/Argentinean sector. The drifting of the APV during the spring has been studied over the past decade (Hassler et al, 2011, Grytsai et al., 2017) using both observations




and models. The results show longitudinal displacements of the APV across the years. In particular, the minimum of the planetary wavenumber 1, centred at latitudes of approximately 65° S, has shifted westward since 2003, thus increasing the time of the stations in the Argentinean sector under the influence of the PVA. To confirm this effect over Ushuaia and Marambio, the $NO_2$ representative potential temperature level of 530 K between days 150 and 330 (May through November) has been used to calculate the position of the station with respect to the vortex according to the widely accepted Nash et al.

criteria (Nash et al., 1996), based essentially in finding the maximum PV gradient at equivalent latitudes to define the vortex edge. ERA-Interim data were used for this purpose. The results show an increasing number of days inside the vortex both in Ushuaia and Marambio in previous years (Fig. 9). As a consequence, we attribute the opposite behaviour of the winter trends at Ushuaia and Marambio to the slight drift/increase in the vortex position towards the South American sector in the past few years.

The trends at Belgrano in the summer are lower than those at Marambio, with values close to those at Ushuaia. The winter data are not significant. The $NO_2$ concentration in the stratosphere during these months is very low and very dependent on the time of the last warming, which is highly variably from year to year.

**4.2 The subtropical case: comparisons with other observations**

DOAS trends at Izaña have been compared with the ground-based NDACC FTIR data located in the same station and with 3

satellite instruments: nadir SCIAMACHY (SCanning Imaging Absorption spectroMeter for Atmospheric CHartographY) and OMI (Ozone Monitoring Instrument), both DOAS spectrometers, and the limb sounding FTIR MIPAS (Michelson Interferometer for Passive Atmospheric Sounding). The SCIAMACHY stratospheric data are produced by the IUP-Bremen. The data used here are the improved version V3.0 released in 2016. The MIPAS $NO_2$ profiles are from (IMK/IAA data version v5h NO2 20/v5r NO2 220) at the station's location converted to VCD. Only the daytime data (10 AM overpasses) were

considered. The MIPAS data have been corrected with the DOAS averaging kernels to make them comparable. The basic instruments are shown in Table 4. Further details can be found in the corresponding web pages. Figure 10 shows the annual mean values (upper panel) obtained from the monthly means (lower panel) to avoid potential biases due to a non-uniform distribution of data available across the year. This is particularly important for the ground-based FTIR data, since there are many more daily measurements during the summer months, when $NO_2$ is at maximum, than in other months. Only in the few

cases when the monthly data were not available were the corresponding climatological values used. No photochemical correction has been applied to the satellite data to refer them to the same SZA because the purpose is to compare the $NO_2$ column trends. The differences between the instruments for the annual means are within 5 % and exhibit similar seasonalities (Fig. 10, lower panel). There are multiple causes affecting the magnitude of the measured column, namely, the hour of the measurements, field of view, vertical sensitivity, etc. (Piters et al., 2011, Robles-Gonzalez et al., 2016); however, these factors

should not affect the observed trends. Table 5 displays the results of the trends from the regression model using the monthly data. There is a large scatter in the trends. The MIPAS decadal trend is of +3.0 ± 0.4 %, close to that of FTIR (+3.6 ± 2.7 %). SCIAMACHY provides a low trend but is statistically non-significant. OMI shows a decadal trend of +7.5 ± 2.2 %, not far



from that obtained from DOAS but for the 2002-2014 period. MIPAS and OMI are significant at the 90% confidence level. In summary, at present, there is no agreement between the instruments and the actual trend over Izaña. To further complicate

the interpretation, McLinden et al., (2001), using a 3-D CMT model, found that the trend can be highly dependent on the time of day of the measurement. For the case shown in their paper, i.e., with decreasing $O_3$ and increasing halogens, there is almost a factor of 2 between the maximum trend at sunrise and the trend minimum at sunset. In the late morning and around noon, when the satellites and FTIR measurements are taken, the trend is much lower than that at SZA=90° AM, when the DOAS takes its measurement. Therefore, unless careful analyses of the CTM models are carried out for the exact periods of the

measurements, the trend intercomparisons of a set of measurements obtained at different times of day has been found to be of little use for the confirmation of the DOAS trend. There is little doubt, however, of the sign of the trends since all instruments show a positive trend.

**4.3 Discussion of the hemispheric asymmetry**

The opposite trends observed at the NH and SH stations provide evidence that $N_2O$ oxidation is not the cause of the observed

trend, nor of other global parameter changes. Stratospheric temperatures from the ECMWF ERA-Interim data above the NH and SH stations show no statistically significant trends during the period of observation. The EHF has been used as a proxy for the meridional transport as a possible explanatory variable for the trends observed in the analysis, but no significant correlation has been obtained. Eckert et al., (2014) found a similar hemispherically asymmetric pattern when analysing the global ozone trends from MIPAS. The trends with negative signs were found in the northern lower stratosphere, whereas

positive values were observed in the Southern Hemisphere. These authors suggested a meridional displacement of the subtropical barriers as the cause of the hemispheric asymmetry, since they could mimic the observed trends by shifting the subtropical mixing barriers to the south by 5° at altitudes below 30 km, although no explanation was provided for the displacement beyond what is possible due to low-frequency natural variations. Previously, Stiller et al., (2012) computed the age of stratospheric air (AoA) in the stratosphere using MIPAS sulphur hexafluoride ($SF_6$) measurements, finding a positive

trend in the Northern Hemisphere, centred at 20 km from 20º to 60° N and a negative trend in the tropics and the Southern Hemisphere. In particular, the increasing AoA was observed at heights from the tropopause to the upper limit of MIPAS (38 km). The MIPAS AoA has been reassessed (Haenel et al., 2015), but the hemispheric asymmetry remains. Total positive reactive $NO_y$ trends in the Northern Hemisphere and the negative trends in the Southern Hemisphere in the lower stratosphere (~25 km) have been reported by Funke et al., (2015), using the MIPAS data from the period 2002-2012. A more detailed

meridional structure of the $NO_2$ trends was obtained by Burrows et al., (2016) by analysing the SCIAMACHY $NO_2$ limb measurements in 2003-2011. They find a well-defined region with positive/negative trends extending from the equator to the high latitudes in the Northern/Southern Hemispheres, with a maximum at 26-27 km in the subtropics.

Since the global $NO_y$ remained almost constant during this period, the observed behaviour should be the result of a meridional redistribution. Recently, Stiller et al., (2017) found the same meridional pattern in $N_2O$ from the CLaMS CTM

model, driven by the ERA-Interim reanalysis, and explored the hemispheric symmetry in the AoA, confirming a southward



displacement of the stratospheric circulation pattern between the potential temperature levels of 500 and 800 K. Additionally, Garfinkel et al., (2017), using the GEOSCCM model, also found an increase in the Northern Hemisphere AoA for the period after 1992 when including the stratospheric aerosols and ODS evolution in the simulation. The positive trend peaks between 20-30° N and extends from 20 to 40 km.

390       In summary, previous works exhibit discrepancies in the heights where the stratospheric trends peak (AoA, ozone and $NO_2$) but agree on the existence of a change of sign in the stratospheric trend in both hemispheres. Although the search for an explanation of the observed hemispheric asymmetry patterns is outside the scope of this paper, there is growing evidence based on observational analysis and modelling of a redistribution of tracers in the stratosphere in the past few decades. The results presented here provide an additional confirmation of the changes in the dynamics of the lower/middle stratosphere

based on data from an independent source.

## 5 Conclusions

Long-term datasets from the DOAS spectrometers located at remote stations (Izaña, 28° N, Ushuaia, 55° S, Marambio, 64° S and Belgrano, 78° S) have been used to calculate the trends in the $NO_2$ stratospheric column over the 1993/94-2014 period. Seasonal cycles, stratospheric aerosols, solar cycle, ENSO, QBO, stratospheric temperature, and eddy heat flux were included

as explanatory variables. The results show a positive mean $NO_2$ decadal trend of 8.7 ± 1.2 % over the past 23 years at the northern subtropical station at Izaña, surpassing the expected 2.5 % increase due to the positive $N_2O$ trend. The observed trend is larger at dawn (9.5 ± 1.2 %) than at dusk (7.9 ± 1.1 %). The attempt to compare other available databases does not provide confirmation of the observed DOAS trend, either due to the shorter time series, larger uncertainties, or different diurnal sampling. The same analysis applied to the Antarctic/Subantarctic stations displays the opposite trends. The $NO_2$ stratospheric

column has been found to decrease at a mean rate of -9.1 ± 1.2 %.decade$^{-1}$ and -13.8 ± 2.2 %.decade$^{-1}$ for Ushuaia and Marambio, respectively. Most of the observed trends occurred in the last decade (after 2003-2006). Belgrano also displays a negative trend of -11.3 ± 4.0 % for the summer/autumn season, which is the only period that is statistically significant. The trend in the $NO_2$ diurnal build up (DBU) rate is essentially due to $N_2O_5$ photolysis and is +5.5 ± 1.6 % at Izaña, -8.4 ± 1.8 % at Ushuaia and -13.8 ± 2.4 % at Marambio, providing a hint about the $NO_y$ trend.

410       The analysis of individual months shows that the trend is largely dependent on the season. The decadal trends annual excursion at Izaña ranges from 4.4 ± 1.8 % at its minimum in February to 13.8 ± 2.8 % in December. The Southern Hemisphere station trends are strongly influenced by the Antarctic polar vortex. The negative trend at Marambio increases in mid-winter (-21 %), whereas at Ushuaia, the decadal negative trend is reduced to values close to its annual minimum during the same time (-7 %). The largest trend is observed in September. The considerable difference in the trends can be explained by a shift

of the APV towards the South America sector during the winter. The analysis of the trends, given the position of the stations with respect of the Antarctic polar vortex (APV) over the Antarctic region, supports this explanation.





The results presented here provide an additional confirmation of the changes in the lower/middle stratosphere dynamics based on data from an independent source and provide further observational evidence of the recent findings on the hemispheric asymmetry in stratospheric $NO_2$ and the age of air during the past decades.

**6 Acknowledgements**

The authors want to acknowledge the station operations teams, who were particularly valuable at the remote locations. The FTIR data were provided by the Izaña-FTIR team at the Karlsruhe Institute of Technology (Thomas Blumenstock). Long-term measurements were made possible thanks to the funding provided by the UE Framework programme NORS (FP7/2007-2013 under grant agreement n°284421) and Spanish R+D Plan projects AMISOC (CGL2011-24891), AVATAR (CGL2014-
55230-R), VIOLIN (CGL2010-20353) and HELADO (CTM2013-41311-P). The trend analysis was programmed with the free software GNU OCTAVE.

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



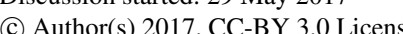

**Figure 1: NO₂ vertical column density time series (blue lines) and fit obtained using the multiple linear regression model (red lines)**





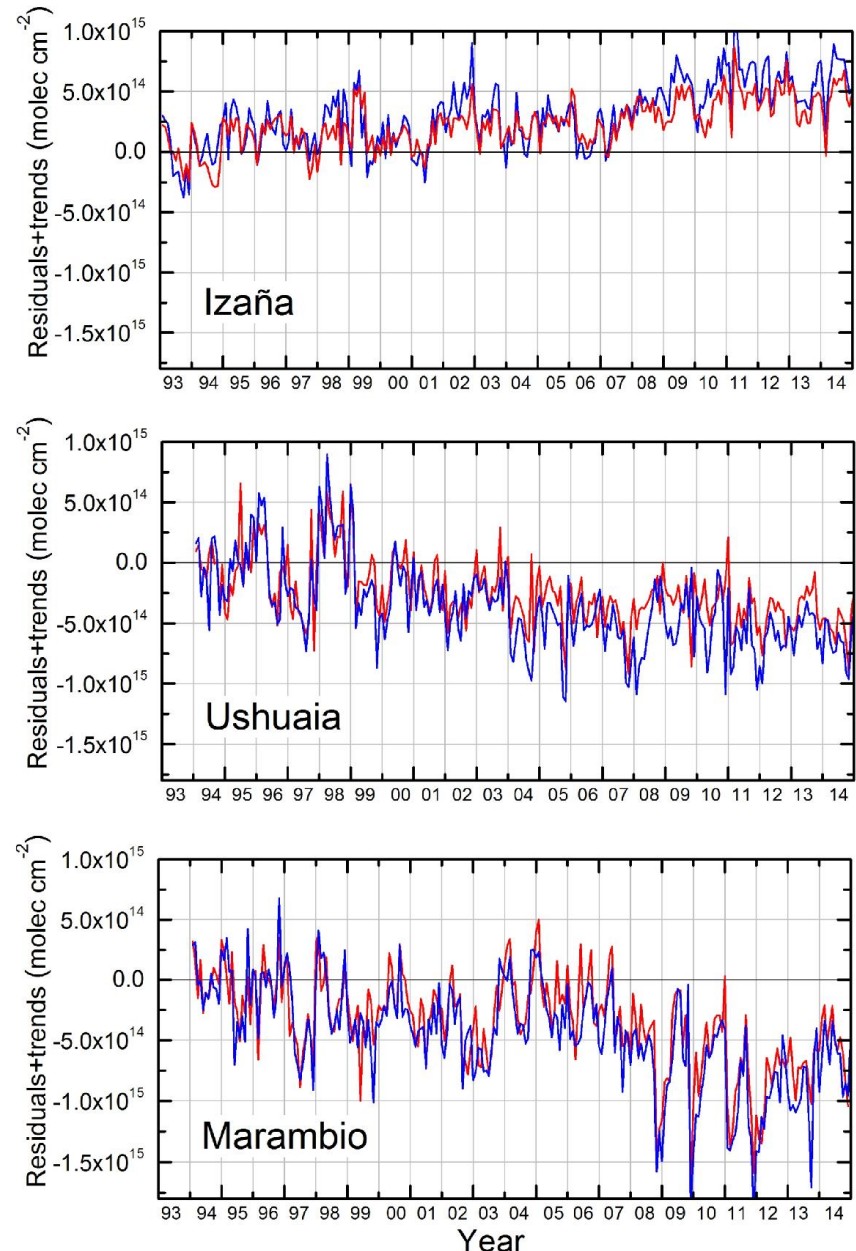

**Figure 2. Fit residuals: observations minus models without trends. Red lines: AM data. Blue lines: PM data.**






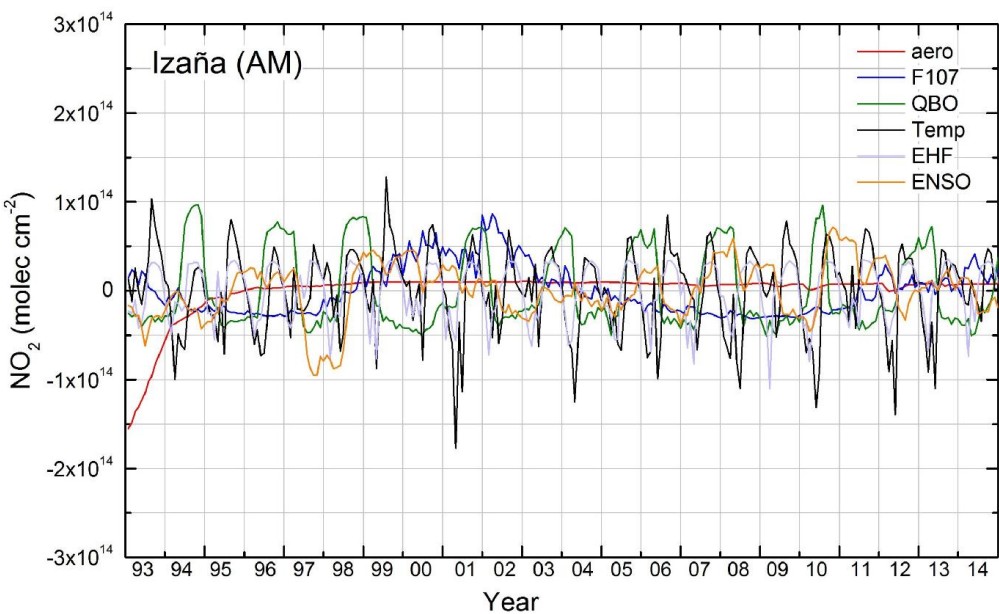

**Figure 3. Example of the contributions of the proxies other than the seasonality and trends to the modeled NO₂ time series for Izaña.**

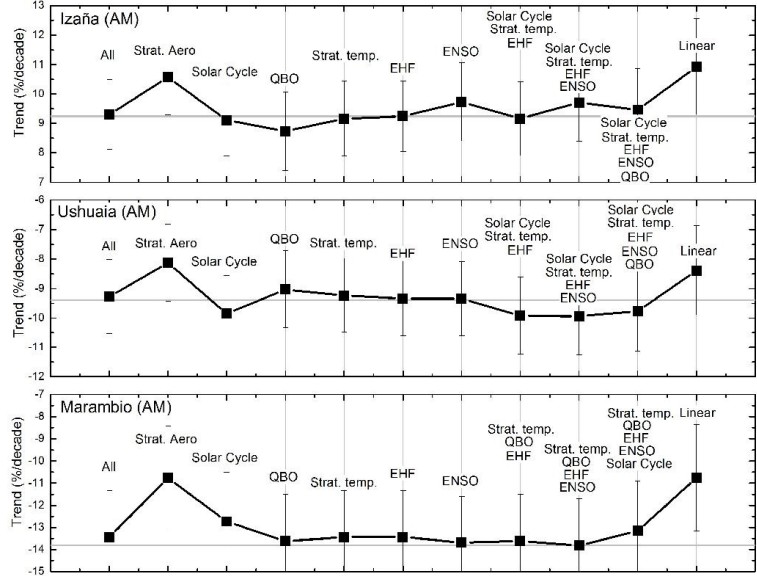

**Figure 4. Impact of the computed trends per month depending on the selected proxies. Case "All" = all proxies included. Labels indicate the proxy/ies excluded from the analysis. Trend obtained in the final analysis is plotted as a reference (grey line). See text for details.**




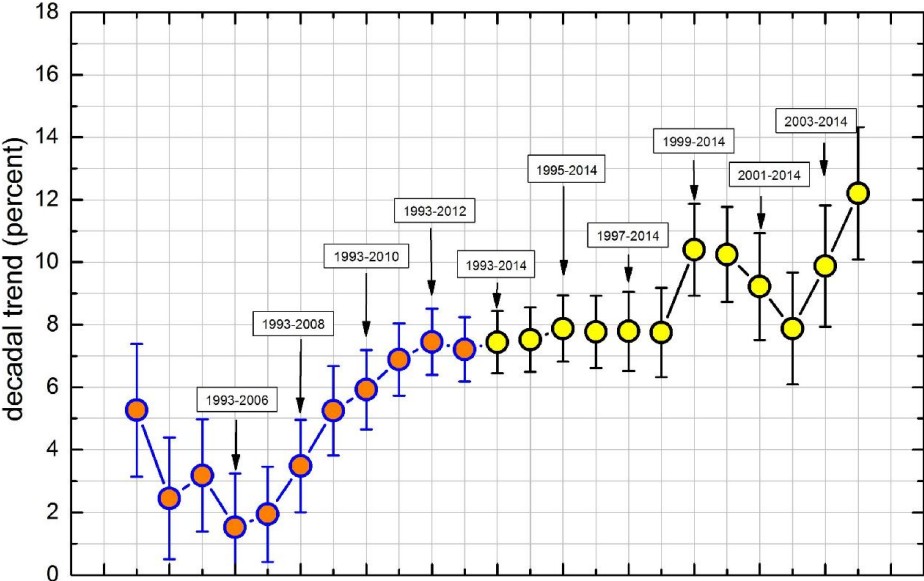

**Figure 5. Dependency of the Izaña PM trend on the selected period. In the central point, the complete 1993-2014 time series is used. Each point to the left (orange circles) is the trend after reducing the time series by one year before 2014. Each point to the right (yellow circles) is the trend after reducing the time series by one year after 1993.**


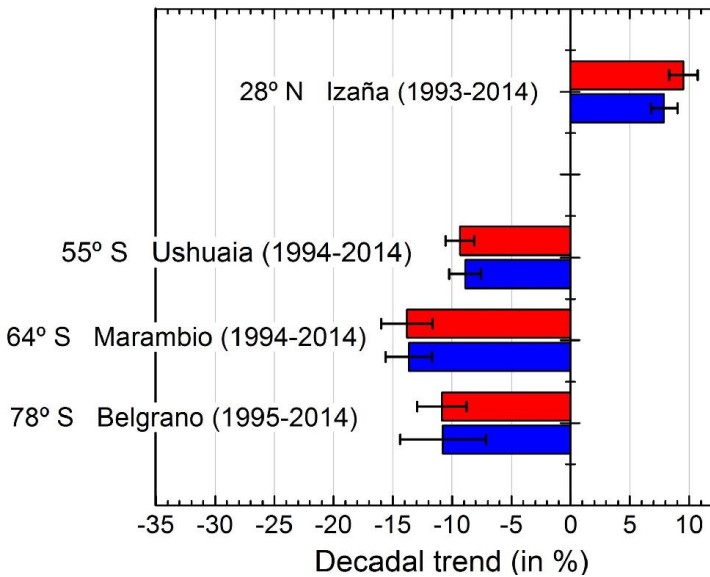

**Figure 6. AM (red) and PM (blue) decadal trends obtained for the DOAS stations. The trend at Belgrano was obtained for the Feb-March-April period. See the text for details.**





**Figure 7. Diurnal NO₂ build-up in molec.cm⁻².h⁻¹ for the 4 DOAS stations. a)-c) Observed DBU (black lines). Annual means (red lines). Linear trends (dotted lines). The computed DBU is plotted in a) (blue line). d) The Belgrano DBU for all available months (black stars). Months of March and April are highlighted with red circles and blue squares, respectively.**





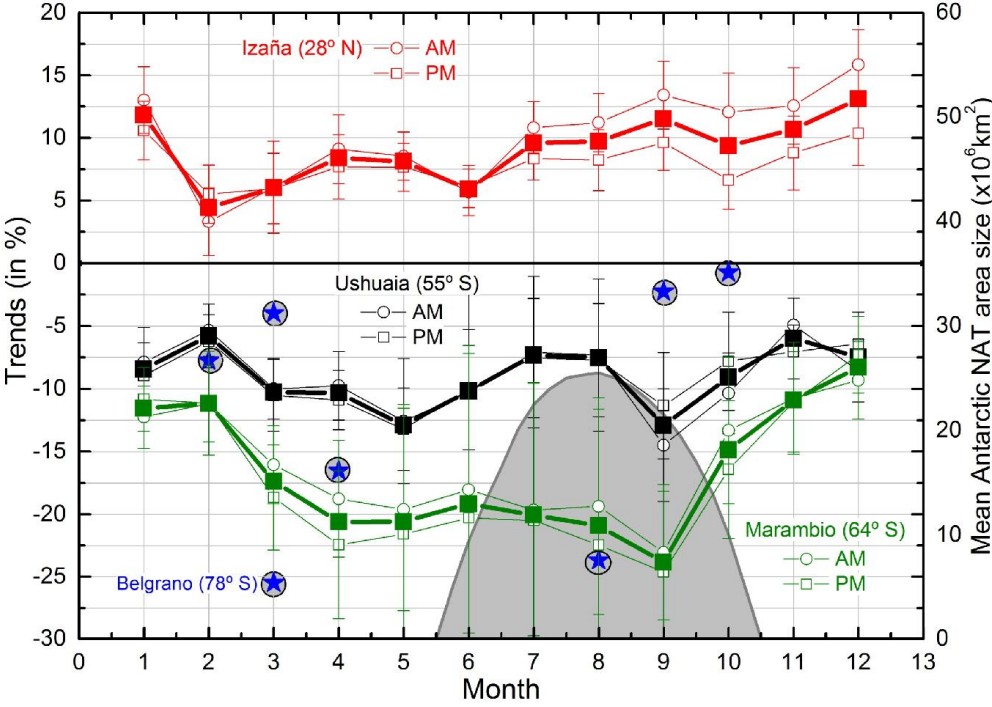

**Figure 8. NO₂ decadal trends for the individual months at Izaña, Ushuaia, Marambio and Belgrano. Open circles and open squares represent the AM and PM data, respectively. Solid squares are the monthly means. The Belgrano trends are from the diurnal means. The shadowed area is the mean size of the area of the potential NAT PSCs (right scale).**

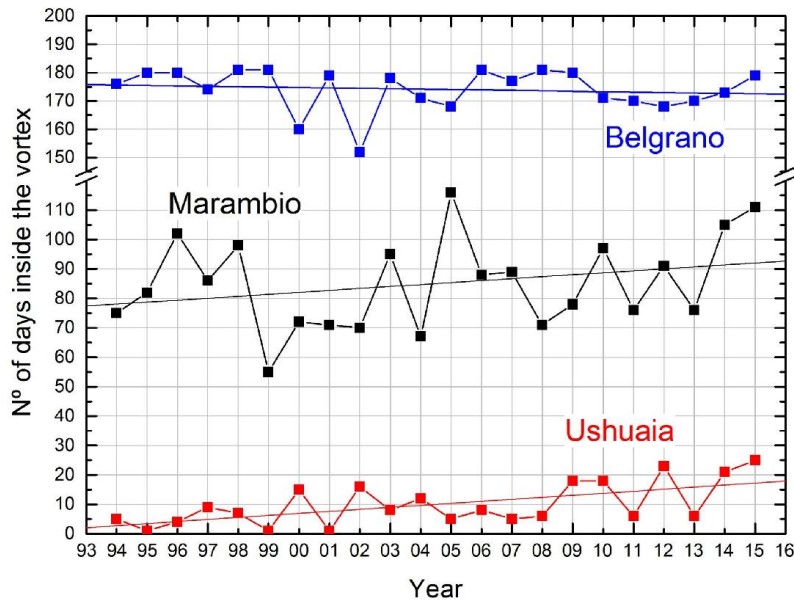

**Figure 9. Annual evolution of the number of days that the stations spent inside the polar vortex at 530K according to the Nash et al. (1996) criteria.**





**Figure 10. NO₂ VCD annual means at Izaña from the available datasets (upper panel) obtained as monthly averages (lower panel).**




**Table 1. DOAS retrieval settings**.

|  | Izaña: EVA/ RASAS / RASAS-II | EVA: Ushuaia/Marambio/Belgrano |
|---|---|---|
| Fitting interval | EVA: 430-450 nm<br>RASAS-I: 450-533 nm<br>RASAS-II: 430-520 nm | 430-450 nm |
| $NO_2$ cross-section | Vandaele et al. (1998), 220K | Vandaele et al. (1998), 220K |
| $O_3$ cross-section | Bogumil et al. (2001), 223K | Bogumil et al. (2001), 223K |
| $H_2O$ cross-section | Hitran (Rothman et al., 2008) | Hitran (Rothman et al., 2008) |
| $O_4$ cross-section | EVA/RASAS. Greenblatt (1990), room temp.<br>RASAS-II. Hermans (1999), room temp. | Greenblatt (1990), room temp. |
| Ring effect | Chance and Spurr (1997) | Chance and Spurr (1997) |
| Orthogonalization Polynomial | EVA: 2nd degree<br>RASAS/RASAS-II: 3rd degree | 2nd degree |
| Offset correction | Inverse of the reference | Inverse of the reference |
| AMF calculation | NDACC $NO_2$ AMF LUTs | NDACC $NO_2$ AMF LUTs |
| Determination of the residual amount in the reference spectrum | Modified Langley plot (Vaughan et al., 1997) | Modified Langley plot (Vaughan et al., 1997) |
| SZA range for the twilight averaging of the vertical columns | EVA: 88 – 91° SZA<br>RASAS/RASASII: 89 – 91° SZA<br>(Approx. 6 measurements) | 88 – 91° SZA<br>(Approx. 6 measurements) |









**Table 2. Source of the explanatory variables of the multiple regression.**

| Description | Source |
|---|---|
| Stratospheric aerosols | http://data.giss.nasa.gov/modelforce/strataer/ |
| Solar cycle | http://umbra.nascom.nasa.gov/sdb/ydb/indices_flux_raw/Penticton_Observed/monthly/MONTHPLT.OBS. |
| Quasi Biennial Oscillation (50 and 10 hPa) | http://www.geo.fu-berlin.de/met/ag/strat/produkte/qbo/qbo.dat. |
| El Niño Southern Oscillation | http://www.esrl.noaa.gov/psd/enso/mei/table.html. |
| Strat. temperature and wind for the Eddy heat flux calculation | http://apps.ecmwf.int/datasets/data/interim-full-daily |









**Table 3. Summary of the trend analysis for the DOAS stations. In the lower part, the contribution of each explanatory variable to the total variance is given in %. The colors denote the significance (blue 99%, pink 95% and green 90%). "Not" means it has been excluded from the definite analysis**.

| | IZAÑA (28° N) | | USHUAIA (55° S) | | MARAMBIO (64° S) | |
|---|---|---|---|---|---|---|
| | AM | PM | AM | PM | AM | PM |
| Trend (in %) | +9.51 | +7.89 | -9.34 | -8.91 | -13.94 | -13.64 |
| Uncertainty (in %) | 1.19 | 1.11 | 1.27 | 1.32 | 2.09 | 1.95 |
| Uncertainty (in mol.cm$^{-2}$.decade$^{-1}$) | $2.61 \times 10^{13}$ | $3.79 \times 10^{13}$ | $3.35 \times 10^{13}$ | $4.98 \times 10^{13}$ | $5.95 \times 10^{13}$ | $6.75 \times 10^{13}$ |
| Autocorrelation coeff. | 0.478 | 0.525 | 0.275 | 0.487 | 0.457 | 0.555 |
| Residual Standard dev. | $1.53 \times 10^{14}$ | $2.08 \times 10^{14}$ | $2.42 \times 10^{14}$ | $2.80 \times 10^{14}$ | $3.50 \times 10^{14}$ | $3.43 \times 10^{14}$ |
| **Contribution to the variance and confidence level (in %)** | | | | | | |
| Trend | 27.82 | 26.96 | 9.90 | 10.21 | 8.50 | 10.47 |
| Annual wave | 56.61 | 55.59 | 81.11 | 81.96 | 80.74 | 84.37 |
| Semiannual wave | Not | Not | 0.56 | 0.41 | 6.09 | 1.08 |
| Quarterly wave | Not | Not | Not | Not | Not | Not |
| Strat. Aerosols | 0.29 | 0.75 | 0.17 | 0.16 | 0.34 | 0.24 |
| Solar Cycle | 0.24 | 0.01 | 0.25 | 0.03 | 0.18 | 0.20 |
| QBO | 0.51 | 0.95 | 0.29 | 0.11 | Not | Not |
| Strat. Temperature | 0.95 | 1.19 | 1.83 | 2.48 | Not | Not |
| EHF | 0.00 | 1.28 | Not | Not | Not | Not |
| ENSO | 0.41 | 0.08 | 0.01 | 0.10 | Not | Not |
| TOTAL variance explained (%) | 86.83 | 86.81 | 94.10 | 95.47 | 95.84 | 96.37 |
| Contributions other than those of season and trend (%) | 2.40 | 4.26 | 2.55 | 2.88 | 0.52 | 0.44 |

**99%** [ ]   **95%** [ ]   **90%** [ ]







**Table 4. Instrument information.**


| Instrument | Platform | Technique | Data | Period | Data availability |
|---|---|---|---|---|---|
| FTIR | Ground-based | FTIR | | 1/2000-12/2013 | IMK/ASF NDACC-Izaña FTIR team https://www.imk-asf.kit.edu/english/201.php |
| MIPAS | ENVISAT | FTIR | | 7/2002-4/2012 | IMK/IAA MIPAS teamhttp://www.imk-asf.kit.edu/english/308.php |
| SCIAMACHY | ENVISAT | DOAS | | 8/2002-3/2011 | http://www.iup.uni-bremen.de/doas/scia_no2_data_acve.htm |
| OMI | AURA | DOAS | V3.0 | 1/2005-12/2014 | http://avdc.gsfc.nasa.gov/pub/most popular/overpass/OMI/OMNO2/ |


**Table 5. Decadal trends in available datasets at Izaña.**

| Technique | Measurements | Period | Total months | Gaps | Trend %/decade | Error %/decade | Significance 90% | Residual Standard Dev | T Statist. | SA | SC | TEM | ENS | NAO | QBO | EHF |
|---|---|---|---|---|---|---|---|---|---|---|---|---|---|---|---|---|
| **DOAS** | AM(SZA=89-91°) | 3/1993-12/2014 | 264 | 9 | + 9.24 | ± 1.20 | YES | $1.51 \times 10^{14}$ | + 11.64 | x | x | x | x | | x | |
| | PM(SZA=89-91°) | 3/1993-12/2014 | 264 | 9 | + 7.53 | ± 1.12 | YES | $2.06 \times 10^{14}$ | + 11.71 | x | x | x | x | | x | x |
| | Diurnal build-up/hour | 1/1993-12/2014 | 264 | 12 | + 5.46 | ± 1.67 | YES | $9.54 \times 10^{12}$ | + 4.65 | x | x | | | | x | |
| **SCIAMACHY** | ≈ noon | 8/2002-3/2011 | 116 | 0 | + 1.34 | ± 2.25 | NO | $1.09 \times 10^{14}$ | + 0.72 | | x | | x | | | |
| **OMI** | ≈ noon | 10/2004-12/2014 | 123 | 0 | + 7.53 | ± 2.24 | YES | $1.39 \times 10^{14}$ | + 4.76 | | | | x | | x | |
| **FTIR** | SZA<60° | 1/2000-12/2013 | 168 | 21 | + 3.61 | ± 2.73 | NO | $2.37 \times 10^{14}$ | - 0.14 | x | x | x | | | | |
| **MIPAS** | ≈ 10 am | 7/2002-4/2012 | 119 | 25 | + 5.02 | ± 3.54 | NO | $1.20 \times 10^{14}$ | +1.42 | x | | | x | | x | |
