# Peer review of "Hemispheric asymmetry in stratospheric NO2 trends"

_Atmospheric Chemistry and Physics, 2017_

## Referee Comment (RC1) · Anonymous Referee #1 · 21 Jul 2017

In this paper, the authors report on long-term ground based zenith-sky observations of stratospheric $NO_2$ columns at four stations and how they changed over the last 20 years. Interestingly, trends are of the order of 10% per decade at all stations but have opposite signs for the NH sub-tropical station and for the SH high latitude stations. The stability of the trends with respect to time interval and parameters included in the trend model is evaluated, the difference between AM and PM trends discussed and the change in diurnal build-up of $NO_2$ evaluated. For Izana, DOAS $NO_2$-trends are also compared to other data sources from FTIR and satellite observations and qualitative but not quantitative agreement is found.

In my opinion, this is an interesting and thorough analysis of stratospheric $NO_2$ trends

which provides interesting data and results. While interpretation is mostly limited to qualitative discussion, and detailed comparison to dedicated model runs for the locations of the instruments is needed for a more quantitative evaluation, the study is relevant enough as it is to warrant publication. The manuscript is clearly structured, well written and the figures are clear and useful. As I have only few comments and suggestions, I recommend publication of this paper after minor revisions.

**Major Comments**

In the discussion of the difference in $NO_2$ trends between Marambio and Ushuaia, it is stated that the months with large differences are linked to the presence of the polar vortex. While this sounds plausible, it is not really supported by Fig. 8 which shows large differences already in April. To me this appears more like a latitudinal dependence which is more pronounced in winter than in summer.

It is also stated that the trend in $NO_2$ columns could be linked to changes in vortex position and the resulting change in statistics. This makes sense, and at least for Ushuaia, it would be relatively simple to check this assumption by repeating the analysis but excluding all measurements where the station was influenced by polar vortex air. I'd suggest adding this analysis to the paper.

In the comparison to satellite data, it would be good to add some information on collocation criteria used and the respective overpass times of the satellites. I think it would also be interesting to compare the satellite data to the PM DOAS measurements. Although the time difference between satellite observations and AM data is usually shorter, the AM measurements are strongly impacted by night-time chemistry whereas the satellite data at least over sub-tropical regions are more representative for daytime chemistry. In my experience, correlation of ground-based and satellite $NO_2$ data is better when using PM observations at least at low latitudes.

**Minor comments**

Line 38: I think the reactions listed do not lead to catalytic ozone destruction; for this reaction of $NO_2$ with O needs to be included

Line 56: major source of $NO_2$ => major source of $NO_2$ in the stratosphere

Line 93: was installed Antarctica => was installed in Antarctica

Line 116: were derived from 6 typical individual measurements => were typically derived from 6 individual measurements (?)

Line 134: effect on the cross-sections => effect on the effective cross-sections

Line 141: For monthly data – do you mean the fraction of months for which you have no data at all?

Line 160: alpha > 0.1 – alpha not defined

Line 160: significant values of less than 90% => significance values of less than 90%

Line 266: Both halogens should result => The observed changes in both halogens should result

Line 192: thus there was less $N_2O_5$ – while this is a reasonable explanation, I think other explanations cannot be completely excluded

Line 364: $N_2O$ oxidation is not the cause of the observed trend, nor of other global parameter changes. – this sentence sounds odd, please rephrase

Figure 10: Please add that these are AM DOAS measurements (see also my comment above).

---

## Referee Comment (RC2) · Anonymous Referee #2 · 8 Aug 2017

This paper presents sound analysis of four data series that, in keeping with NDACC requirements, have measured stratospheric species to the best available standards over decades. The literature is well surveyed, and the analysis techniques follow established practice, but in doing so the paper does not add much new insight. The results are at odds with cited previous studies, and that should be discussed further.

My main criticism is that, starting with the title, the representativeness of the sites seems to be exaggerated; one site in the Canary Islands serves as a proxy for the northern hemisphere, while three sites in Tierra del Fuego, the Antarctic Peninsula, and the southern Weddell Sea purportedly represent the southern hemisphere. It would be better to acknowledge the respective limitations, especially as the NDACC as a whole gives much wider (if still irregular) coverage. Both latitudinal and longitudinal gradients

may be present, as implicit for Antarctic vortex displacement toward South America, but also cited as a feature of Gruzdev's 23-station analysis in 2009 ("Trends were found to be mostly positive in the middle and low latitudes of the SH and mostly negative in the European sector of the middle latitudes of the NH" - both different from this study).

It may also be that the difference depends heavily on the analysis period. The Izaña residuals in Fig 2 to 2008, as comparable to a 2009 analysis, show no upward trend, and the Ushuaia and Marambio data to the same time show indistinct trends.

On the subject of time periods, the authors should perhaps also mention why their paper in mid 2017 only includes data to the end of 2014. My guess is that it represents a lengthy period of trying different analyses in what is routinely a frustrating attempt to tease meaning from such datasets, but it would still be better for the 25 years of NDACC special issue to use the 24 years of data from Izaña and 23 from the others.

Specific points:

lines 41-43: The technique of zenith-sky DOAS was pioneered by Noxon, and the first regular monitoring was at Lauder, continuous from 1980, and later from Scott Base.

line 106: "... degradation ... at a rate of 4.33%/year." Confidence limits on this figure are relevant to the inferred trends.

line 147: The second beta terms should be subscripted 2k, not 2k-1.

line 154: "... since the start of measurements, in months". The scaling of t by 2pi/12 in the trigonometric terms of the equation means that t must be expressed in months.

line 156: "annual, semi-annual, and four-monthly waves". A quarterly term would have k=4.

line 160: "alpha > 0.1" is less conventional than p > 0.1 to express significance level.

line 178: "in the first few years of the time series, strongly affected by Pinatubo, which would otherwise bias the trend" would make it clearer that the bias was a genuine

geophysical effect rather than a measurement or statistical artefact.

line 197: "negative trend (Schwarzkopf and Ramaswamy, 2006)", rather than "i.e." Alternatively, "e.g." might have been meant here.

line 244: "large nitrate aerosols load injected following Pinatubo's eruption". Pinatubo injected 20 Mt of sulphur dioxide, which was oxidised to sulphate aerosol over a few weeks. There was no large injection of nitrate aerosols by Pinatubo, or following it.

line 247: "quarterly" is "four-monthly".

line 265: "negative decadal trend of -5.9 $\pm$ 1.5%" would be clearer, without any risk of tautology.

lines 303-306: "The negative trend in the SH increases ... At Ushuaia, the negative trend is reduced to values close to the annual minimum (-7%), whereas in Marambio, the trend increases with respect to previous months". Talking of minima, and reductions, in negative trends is unnecessarily ambiguous. While the reader can perhaps correctly assume a reduction in the absolute value of a negative trend is meant, it is better to choose different words: "The downward trends in the SH become more negative further south ... At Ushuaia, the trend is less negative ..."

line 340: "The MIPAS data have been combined with the DOAS averaging kernels ...", or "converted", rather than "corrected".

line 348: Fig 10, lower panel, does not show similar seasonalities; that would require a monthly plot like Fig 8.

line 351: "MIPAS decadal trend of +3.0 $\pm$ 0.4%" appears as 5.0 $\pm$ 3.5 in Table 5. Please check all such figures.

line 364: "... evidence that N2O oxidation is not the cause ..." Please clarify. Is that argument that N2O is sufficiently well mixed that any change would expect both hemispheres the same?

[Figure]

line 369: "Negative trends were found in the lower stratosphere ..."

line 391: "... change of sign between hemispheres in the stratospheric trend" would be better, except for comments above about whether the four sites really represent their respective hemispheres.

In Table 5, please check the calculation of the "T statistic" (usually "t statistic") as the values appear to be at odds with the trends and their standard errors.
* * *

---

## Author Comment (AC1) · 18 Sep 2017

**Reply to Interactive comments from reviewer #1**

We strongly appreciate the reviewer for his/her helpful and supportive comments. For clarity, our responses to the reviewer comments are in blue.

**Anonymous Referee #1**

Review of "Hemispheric asymmetry in stratospheric NO2 trends" by M. Yela et al. In this paper, the authors report on long-term ground based zenith-sky observations of stratospheric NO2 columns at four stations and how they changed over the last 20 years. Interestingly, trends are of the order of 10% per decade at all stations but have opposite signs for the NH sub-tropical station and for the SH high latitude stations.

The stability of the trends with respect to time interval and parameters included in the trend model is evaluated, the difference between AM and PM trends discussed and the change in diurnal build-up of NO2 evaluated. For Izana, DOAS NO2-trends are also compared to other data sources from FTIR and satellite observations and qualitative but not quantitative agreement is found. In my opinion, this is an interesting and thorough analysis of stratospheric NO2 trends which provides interesting data and results. While interpretation is mostly limited to qualitative discussion, and detailed comparison to dedicated model runs for the locations of the instruments is needed for a more quantitative evaluation, the study is relevant enough as it is to warrant publication. The manuscript is clearly structured, well written and the figures are clear and useful. As I have only few comments and suggestions, I recommend publication of this paper after minor revisions.

**Major Comments**

In the discussion of the difference in NO2 trends between Marambio and Ushuaia, it is stated that the months with large differences are linked to the presence of the polar vortex. While this sounds plausible, it is not really supported by Fig. 8 which shows large differences already in April. To me this appears more like a latitudinal dependence which is more pronounced in winter than in summer. It is also stated that the trend in NO2 columns could be linked to changes in vortex position and the resulting change in statistics.

This makes sense, and at least for Ushuaia, it would be relatively simple to check this assumption by repeating the analysis but excluding all measurements where the station was influenced by polar vortex air. I'd suggest adding this analysis to the paper.

Following the referee suggestion we have excluded the days inside the vortex from Ushuaia data series. Results show that, as expected, the trend decrease slightly (less negative) but the contribution to general trend is very small probably due to the fact that the days inside the vortex at Ushuaia, while increasing along time, are still very small compared with the total number of days. Even excluding these data the wavy patter occurring in winter (reduction of the negative trend in July-August and increase in September-October are mainly due to mid latitude dynamics. As a consequence we have reformulated the text in the following way:

Line 327 after "(Fig. 9)". Removed the sentence starting after (Fig.9).

Line 327 after "(Fig. 9)". Added the following paragraph: "*To test that the impact of the vortex position drifting is the main cause of the wavy structure in the trend observed in the Ushuaia winter (small trends in July-August large ones in September –October, see Fig. 8) we have repeated the monthly trend analysis excluding the days when the station was inside the vortex. Results show that while, as expected, negative trends are reduced, the magnitude is not enough to justify the observed trends. In fact, the changes are of only few tenths of a percent, probably due to the fact that the days inside the vortex at Ushuaia, while increasing along time, are still very small compared with the total number of days. In summary, when only extra-vortex data are used the winter wavy structure remains providing evidence that the evolution of the seasonal $NO_2$ trend is dominated by the mid-latitude dynamics along the year.*"

In the comparison to satellite data, it would be good to add some information on collocation criteria used and the respective overpass times of the satellites. I think it would also be interesting to compare the satellite data to the PM DOAS measurements. Although the time difference between satellite observations and AM data is usually shorter, the AM measurements are strongly impacted by night-time chemistry whereas the satellite data at least over sub-tropical regions are more representative for daytime chemistry. In my experience, correlation of ground-based and satellite NO2 data is better when using PM observations at least at low latitudes.

The purpose of considering all data available, including GB-FTIR and satellite for the Izaña station is to support, through additional and independent measurements, the genuine character of the trend. For that reason in this particular work no collocation has been applied since the absolute magnitude does not affect the trend. On the other hand, at Izaña, the ENVISAT overpassing time coincides with the effective SZA at which AM DOAS measurements are taken thus the photochemical correction has an almost negligible impact in the intercomparisons. OMI overpassing is later, and the $NO_2$ column is some 10% larger due to the diurnal build-up but, as mentioned, our interest lies in the trend rather than in absolute values. Recently a detailed intercomparison at Izaña considering collocation and "effective SZA" have been published (AMT Robles et al. 2016).

**Minor comments**

Line 38: I think the reactions listed do not lead to catalytic ozone destruction; for this reaction of NO2 with O needs to be included.

We have improved the sentence: *Nitrogen oxides interact with ozone both directly and indirectly. Nitric oxide (NO) reacts with ozone, forming $NO_2$ and $O_2$. NO is recovered by $NO_2$ reaction with atomic oxygen and, in day time, by $NO_2$ photolysis. This catalytic reactions result in ozone reduction.*

Line 56: major source of NO2 => major source of NO2 in the stratosphere.

Done

Line 93: was installed Antarctica => was installed in Antarctica.

Done

Line 116: were derived from 6 typical individual measurements => were typically derived from 6 individual measurements (?).

*We have modified the sentence: "…were derived from all available measurements between 89º and 91º SZA, typically 5-6 data per twilight".*

Line 134: effect on the cross-sections => effect on the effective cross-sections.
*Done*

Line 141: For monthly data – do you mean the fraction of months for which you have no data at all?

*We clarified this point: "For monthly mean data, the rates of failures are 3.45 %, 0.40 % and 0.79 % of the total dataset for Izaña, Ushuaia and Marambio, respectively".*

Line 160: alpha > 0.1 – alpha not defined.

*Alpha refers to the significance level. We simplified the sentence "to exclude the proxies exceeding α> 0.1, corresponding to significant values of less than 90%". Now it stands as "to exclude proxies with confidence intervals below 90%"*

Line 160: significant values of less than 90% => significance values of less than 90%.

*Changed. See above*

Line 266: Both halogens should result => The observed changes in both halogens should result.

*Done*

Line 292: thus there was less N2O5 – while this is a reasonable explanation, I think other explanations cannot be completely excluded.

*In lines 277-285 we argue that DBU is a good index to estimate changes in $N_2O_5$. However, as the reviewer points out, other explanations could be possible, particularly a change in the vertical distribution of the $NO_2$ column. Therefore we have modified the sentence as follows:*

*"In summary, all SH stations exhibit a negative decadal trend in their DBUs, ranging from -8 to -14 %, revealing either a reduction of $N_2O_5$ in the past few years at the middle and high latitudes of the Southern hemisphere or dynamically induced changes in the $NO_2$ vertical distribution ".*

Line 364: N2O oxidation is not the cause of the observed trend, nor of other global parameter changes. This sentence sounds odd, please rephrase.

*Global $NO_2$ increase was expected at the beginning of the analysis due to the very long $N_2O$ lifetime at a maintained increasing rate of 2.5%/decade. We have reformulated the sentence to make it clearer:*

*"The opposite sign in the $NO_2$ trends observed at the NH and SH stations shows that the NOx distribution in the stratosphere does not directly reflects the increasing $N_2O$ in the atmosphere, at least when individual stations are analyzed."*

Figure 10: Please add that these are AM DOAS measurements (see also my comment above).

*Done*

NOTE: Ushuaia data in figure 2, 4, 5, 6 and 8 are slightly different than in previous version due to the correction of few data from a recent quality control.

---

## Author Comment (AC2) · 18 Sep 2017

**Reply to Interactive comments from reviewer #2**

We thank you the reviewer for his/her helpful comments. For clarity, our responses to the reviewer comments are in blue.

**Anonymous Referee #2**

This paper presents sound analysis of four data series that, in keeping with NDACC requirements, have measured stratospheric species to the best available standards over decades. The literature is well surveyed, and the analysis techniques follow established practice, but in doing so the paper does not add much new insight. The results are at odds with cited previous studies, and that should be discussed further. My main criticism is that, starting with the title, the representativeness of the sites seems to be exaggerated; one site in the Canary Islands serves as a proxy for the northern hemisphere, while three sites in Tierra del Fuego, the Antarctic Peninsula, and the southern Weddell Sea purportedly represent the southern hemisphere. It would be better to acknowledge the respective limitations, especially as the NDACC as a whole gives much wider (if still irregular) coverage.

While certainly a subtropical station is not representative of the NH, all 4 NDACC GB stable instruments with long data series support the hemispheric asymmetry in stratospheric NO2 at different latitudes previously observed by satellite instrumentation. The title highlight what we think is the major outcome of the paper: The changing behavior of the stratospheric dynamics in last decades. The stations are already enumerated in the abstract, so we think there is no misunderstanding.

Both latitudinal and longitudinal gradients may be present, as implicit for Antarctic vortex displacement toward South America, but also cited as a feature of Gruzdev's 23-station analysis in 2009 ("Trends were found to be mostly positive in the middle and low latitudes of the SH and mostly negative in the European sector of the middle latitudes of the NH" - both different from this study).

Latitudinal and even longitudinal gradients in the tracers distribution in the stratosphere are a well known feature. NO₂ display a strong meridional gradient during solstice due to differences in illumination time. Longitudinal small gradients can be present as well due to the quasi-stationary planetary waves, but we don't see the reason why this should affect the long-term trends. On the other hand Gruzdev data analysis ends in 2007 and we found that last years have a strong contribution to the observed trend.

It may also be that the difference depends heavily on the analysis period. The Izaña residuals in Fig 2 to 2008, as comparable to a 2009 analysis, show no upward trend, and the Ushuaia and Marambio data to the same time show indistinct trends.

Time period determine the magnitude in any analysis trend. In figure 5 the Izaña trend resulting from calculation using different time periods show that the trend remains essentially unchanged if the data series is shortened by up to 5 years at the start and up to 4 at the end, but also that most of the trend resulted from the increase in last decade. A similar situation occurred in the SH. (See Fig.2 and text in lines 255-256).

On the subject of time periods, the authors should perhaps also mention why their paper in mid 2017 only includes data to the end of 2014. My guess is that it represents a lengthy period of trying different analyses in what is routinely a frustrating attempt to tease meaning from such datasets, but it would still be better for the 25 years of NDACC special issue to use the 24 years of data from Izaña and 23 from the others.

The reason why the data series ends at the beginning of 2015 is that the work was mostly elaborated during 2015 and delayed later on due to a number of reasons. Adding 2 more years of data (10%) at the present time implies a complete recalculation that authors do not consider justified enough since it would produce only small changes in the magnitude of the results not compromising the main results of the paper.

**Specific points:**

Lines 41-43: The technique of zenith-sky DOAS was pioneered by Noxon, and the first regular monitoring was at Lauder, continuous from 1980, and later from Scott Base.

We corrected this omission. Authors are well aware that Lauder station is the first regular $NO_2$ monitoring station. The sentence now stands as: *"Regular monitoring of stratospheric $NO_2$ started in 1980 with the deployment of Zenith DOAS (Differential Optical Absorption Spectroscopy) scanning spectrometers at Lauder, New Zealand by Mckenzie and Johnston, 1982 followed by two Antarctic stations (Scott Base at 78° S by Mckenzie and Johnston, 1984 and Dumont D'Urville at 66° S by Pommereau and Goutail, 1988)"*.

We have included reference: McKenzie, R., L. and Johnston, P.V.: Seasonal variations in stratospheric $NO_2$ at 45º S, Geophys. Res. Lett., V9, p. 1255-1258 , 1982

Line 106: "... degradation ... at a rate of 4.33%/year." Confidence limits on this figure are relevant to the inferred trends.

The scanning spectrometer initially installed in 1993 is still in operation. We added the PDA and CCD to improve the accuracy and to extend the spectral range toward the UV and IR. We do not estimate a larger error due to degradation since it was a very slow and linear effect, and was corrected with the scanning-photomultiplier spectrometer operating in the same site (430-450 nm band).

Line 147: The second beta terms should be subscripted 2k, not 2k-1.

Corrected.

Line 154: "... since the start of measurements, in months". The scaling of t by 2pi/12 in the trigonometric terms of the equation means that t must be expressed in months.

Added

Line 156: "annual, semi-annual, and four-monthly waves". A quarterly term would have k=4.

Quarterly term has been removed from the analysis and text since it has no impact at all.

Line 160: "alpha > 0.1" is less conventional than p > 0.1 to express significance level.

We simplified the sentence "*to exclude the proxies exceeding α > 0.1, corresponding to significant values of less than 90%*". Now it stands as "*to exclude proxies with confidence intervals below 90%*"

Line 178: "in the first few years of the time series, strongly affected by Pinatubo, which would otherwise bias the trend" would make it clearer that the bias was a genuine geophysical effect rather than a measurement or statistical artefact.

The sentence has been modified: "SA *proxy accounts for the short-range contribution of the volcanic aerosols on the NO$_2$ column which would, otherwise, affect the long term trend".*

Line 197: "negative trend (Schwarzkopf and Ramaswamy, 2006)", rather than "i.e." Alternatively, "e.g." might have been meant here.

Corrected.

Line 244: "large nitrate aerosols load injected following Pinatubo's eruption". Pinatubo injected 20 Mt of sulphur dioxide, which was oxidised to sulphate aerosol over a few weeks. There was no large injection of nitrate aerosols by Pinatubo, or following it.

This is a miswriting. It is obviously sulphate.

Line 247: "quarterly" is "four-monthly".

The K=3 term does not affect anything and has been removed.

Line 265: "negative decadal trend of -5.9 ± 1.5%" would be clearer, without any risk of tautology.

Changed.

Lines 303-306: "The negative trend in the SH increases ... At Ushuaia, the negative trend is reduced to values close to the annual minimum (-7%), whereas in Marambio, the trend increases with respect to previous months". Talking of minima, and reductions, in negative trends is unnecessarily ambiguous. While the reader can perhaps correctly assume a reduction in the absolute value of a negative trend is meant, it is better to choose different words: "The downward trends in the SH become more negative further south ... At Ushuaia, the trend is less negative ...".

The proposed sentence is clearer. We have modified it.

Line 340: "The MIPAS data have been combined with the DOAS averaging kernels ...", or "converted", rather than "corrected".

Changed

Line 348: Fig 10, lower panel, does not show similar seasonalities; that would require a monthly plot like Fig 8.

*"annual means"* in line 347 should read *"monthly means"*. Lower plot displays the monthly means. This has been corrected.

Line 351: "MIPAS decadal trend of +3.0 ± 0.4%" appears as 5.0 ± 3.5 in Table 5. Please check all such figures.

Figures in Table 5 are correct. We have corrected the text in line 351.

Line 364: "... evidence that $N_2O$ oxidation is not the cause ..." Please clarify. Is that argument that $N_2O$ is sufficiently well mixed that any change would expect both hemispheres the same?.

Global $NO_2$ increase was expected at the beginning of the analysis due to the very long $N_2O$ lifetime at a maintained increasing rate of 2.5%/decade. We have reformulated the sentence to make it clearer: *"The opposite sign in the $NO_2$ trends observed at the NH and SH stations shows that the NOx distribution in the stratosphere does not directly reflects the increasing $N_2O$ in the atmosphere, at least when individual stations are analyzed".*

Line 369: "Negative trends were found in the lower stratosphere ..."

Corrected

Line 391: "... change of sign between hemispheres in the stratospheric trend" would be better, except for comments above about whether the four sites really represent their respective hemispheres.

Corrected.

In Table 5, please check the calculation of the "T statistic" (usually "t statistic") as the values appear to be at odds with the trends and their standard errors.

The stated errors are not the standard errors but the trend estimate errors based on Weatherhead et al, 1998 which account for the data autocorrelation. Since the t-statistics only contributes to generate confusion, we have removed that column from the table.

NOTE: Ushuaia data in figure 2, 4, 5, 6 and 8 are slightly different than in previous version due to the correction of few data from a recent quality control.
.